# Diagnosis and Risk Factors in Retinopathy of Prematurity: A Five-Year Single-Center Descriptive Study

**DOI:** 10.3390/life15091463

**Published:** 2025-09-18

**Authors:** Fatma Sumer, Mehmet Kenan Kanburoglu, Ozgur Altuntas, Fatma Erbatur Uzun, Isil Uslubas, Feyzahan Uzun, Aytac Kanar

**Affiliations:** 1Department of Ophthalmology, Faculty of Medicine, Recep Tayyip Erdogan University, Rize 53100, Turkey; 2Department of Neonatology, Faculty of Medicine, Recep Tayyip Erdogan University, Rize 53100, Turkey; kanburoglumk@outlook.com (M.K.K.);; 3Department of Ophthalmology, Kocaeli City Hospital, Kocaeli 41780, Turkey

**Keywords:** birth weight, gestational age, retinopathy of prematurity, risk factors

## Abstract

Objective: We aimed to determine the incidence and screening outcomes of retinopathy of prematurity (ROP) in preterm infants managed at a tertiary neonatal intensive care unit (NICU) and to identify associated risk factors. Material and Methods: Medical records of 454 premature infants who underwent ROP screening between April 2016 and August 2021 were retrospectively analyzed. Infants with birth weight (BW) ≤ 1500 g or ≤32 weeks of gestational age and those with BW > 1500 g or GA > 32 weeks who had an unstable clinical course were included. All of them were born in the same center. Demographic characteristics, potential risk factors for ROP, ocular examination findings, and treatment requirement were recorded. Results: During the five-year study period, ROP was observed in 75 (16.6%) of a total of 454 premature infants with a mean gestational age (GA) of 30.19 ± 2.49 weeks and a mean BW of 2025.15 ± 614.46 g in the NICU. Of these patients, 67 (14.8%) had stage I disease and 8 (1.8%) had stage II disease. Advanced-stage ROP was not detected in any of the cases. The median GA of patients diagnosed with ROP was 29 weeks (22–35) and the median BW was 2100 g (500–3750), which were significantly lower than those without ROP (*p* < 0.001). When multivariate logistic regression analysis was evaluated with the Wald method, the accuracy rate of the model examining the combined effect of GA, intraventricular hemorrhage (IVH), respiratory distress syndrome (RDS), patent ductus arteriosus (PDA), necrotizing enterocolitis (NEC), and surfactant treatment was 85.9%. In this model, gestational age (OR: 0.712, *p* < 0.001), IVH (OR: 2.915, *p* = 0.010), RDS (OR: 2.129, *p* = 0.004), NEC (OR: 3.679, *p* < 0.001), PDA (OR: 2.434, *p* = 0.021), and surfactant treatment (OR: 2.271, *p* = 0.002) were found to be independent risk factors for ROP development. Conclusions: Small GA and low BW are the main risk factors for the development of ROP. The incidence of ROP was found to be lower than similar studies conducted in our country. While severe ROP cases have been reported in more mature infants in Turkey, our study found no treatment-requiring ROP cases, likely reflecting the higher mean GA and BW characteristics of our cohort.

## 1. Introduction

Retinopathy of prematurity (ROP) is a vasoproliferative vitreoretinopathy characterized by abnormal vascular formations at the junctions of vascularized and avascular retinal regions in premature infants [1]. Despite increased awareness and therapeutic advances, ROP remains a leading cause of childhood blindness, accounting for approximately 10% of cases in developed countries [2]. In Turkey, while comprehensive national blindness registries are limited, available studies indicate that congenital disorders and acquired conditions including ROP constitute major causes of childhood visual impairment [3]. The TR-ROP study, the largest national multicenter study, found that 27% of screened preterm infants developed any stage of ROP, with 6.7% developing severe disease requiring treatment [4]. However, precise national statistics on ROP’s contribution to overall childhood blindness remain unavailable due to the lack of a centralized blindness registry, making ROP prevention and early detection critical public health priorities [5].

The best way to prevent vision loss due to ROP is to establish a good screening program. Even mild stages of ROP reflect arrested retinal vascular development and may lead to long-term complications, including refractive errors, strabismus, amblyopia, reduced visual acuity, and increased risk of retinal detachment in adulthood [6,7]. Studies demonstrate that 30–50% of infants with any stage of ROP develop significant refractive errors, particularly myopia and astigmatism, with 15–25% experiencing strabismus requiring intervention [8,9]. Additionally, long-term follow-up studies demonstrate increased risks of glaucoma, visual field defects, and late-onset retinal complications even in cases of regressed ROP [10,11]. Since all ROP cases in our cohort were mild (stages 1–2), understanding these long-term implications becomes particularly crucial for comprehensive patient counseling and follow-up planning. Therefore, systematic screening and long-term ophthalmological surveillance are essential even for cases that do not require acute treatment. For this reason, each country has established an ROP screening program according to its own conditions [12]. According to the 2018 recommendations of the American Academy of Pediatrics and the American Academy of Ophthalmology, it is recommended to screen all infants born with a birth weight (BW) ≤ 1500 g and/or gestational age (GA) ≤ 30 weeks, as well as infants with a GA greater than 30 weeks and BW between 1500 and 2000 g with clinical problems and requiring cardiopulmonary support [13]. In the UK, ROP screening is recommended for babies born under 1501 g and/or under 32 weeks of gestation [14]. However, some studies suggest that a broader screening program is needed to minimize the risk of blindness caused by ROP. Indeed, Gilbert et al. stated that the criteria in developed countries would not be very suitable for countries with lower socioeconomic development levels and emphasized that the screening limit for these regions should be kept wider and decisions should be made according to local population characteristics [15]. Due to intensive neonatal care conditions in developing countries such as China, the screening program was expanded, and babies born at ≤32 weeks or weighing <1600 g began to be screened for ROP [16]. When the results of the TR-ROP study conducted in our country were evaluated, severe cases of ROP were found in older and more mature infants. As a result of this study, screening of all infants with GA < 34 weeks or BW ≤ 1700 g, or preterm infants with GA ≥ 34 weeks or BW > 1700 g who have been treated with cardiopulmonary support or who are “considered at risk for ROP development by the clinician following the infant”, is recommended [4]. While large multicenter studies like TR-ROP provide essential national epidemiological data, single-center studies provide complementary insights into regional variations in care practices and population characteristics, which are essential for optimizing screening strategies at the institutional level [17].

In this study, we aimed to evaluate the incidence of ROP, clinical outcomes, and associated risk factors in premature infants screened at our NICU over a five-year period, with the goal of developing region-specific screening recommendations.

## 2. Material and Methods

### 2.1. Study Design

Between April 2016 and August 2021, a retrospective cohort study was carried out at the NICU-ROP Screening Clinic at Recep Tayyip Erdoğan University Faculty of Medicine Hospital. Prior to the study, the institutional ethical committee granted approval (IRB No:12.5.2022/116), and all researchers pledged to adhere to the principles of the Declaration of Helsinki.

### 2.2. Data Quality Assurance

To ensure data accuracy and completeness, a systematic data collection and validation protocol was implemented. All patient records were reviewed by two independent researchers, and any discrepancies were resolved through consensus with the senior author. Missing data analysis was conducted to assess potential selection bias, with cases excluded only when critical variables (GA, BW, or ROP examination results) were unavailable.

Data validation included (1) cross-referencing of gestational age and birth weight with delivery records, (2) verification of ROP examination dates and findings against ophthalmology consultation notes, (3) confirmation of comorbidity diagnoses through review of discharge summaries and clinical notes, and (4) double-entry verification for all statistical variables.

Of the initial 472 infants monitored, 18 were excluded for the following reasons:

Major Congenital Anomalies (n = 4): Four infants with life-limiting congenital malformations were excluded, including hypoplastic left heart syndrome (n = 1), anencephaly (n = 1), trisomy 18 (n = 1), and Potter sequence with pulmonary hypoplasia (n = 1). These represent primary developmental disorders unrelated to prematurity that would confound ROP risk factor analysis due to altered systemic physiology and incompatibility with standardized NICU protocols.

Incomplete Retinal Assessment (n = 5): Five infants were discharged or transferred before complete retinal vascularization could be confirmed (mean postmenstrual age of 36 weeks, range 35–38 weeks). Per standard ROP screening guidelines, assessment requires either complete vascularization or follow-up until postmenstrual age of 40 weeks [2]. Early discharge prevented conclusive ROP outcome determination.

Missing Data (n = 9): Nine cases lacked essential documentation, including gestational age (n = 3), birth weight (n = 2), or adequate ophthalmologic records (n = 4). The final study population of 454 infants represents 96.2% of eligible cases, indicating minimal data loss and low risk of selection bias.

### 2.3. Data Collection

Infants with BW ≤ 1500 g or ≤32 weeks GA and those with BW > 1500 g or GA > 32 weeks requiring cardiorespiratory support and who were determined by the attending clinician to be at risk of ROP were screened. Neonates who passed away prior to examination or before the retina achieved full vascularization, as well as infants who did not complete the follow-up, were excluded from the study. A data sheet was prepared for all infants included in the study, documenting gestational age (GA), birth weight (BW), and the presence of any ROP, including severe ROP. In instances where the retina had not been fully vascularized, ophthalmological examinations continued until either full vascularization was achieved or until the infant reached 40 weeks of postmenstrual age. The maximum stage of ROP observed for each infant was recorded, with severe ROP defined as stage 3 or higher according to the International Classification of Retinopathy of Prematurity (ICROP) criteria [18]. The patients included in the study had prenatal characteristics (maternal age, mode of pregnancy (natural or assisted reproduction), presence of eclampsia/preeclampsia, gestational diabetes mellitus (GDM), drug use during pregnancy, premature rupture of membranes, presence of chorioamnionitis, antenatal steroid administration, antepartum hemorrhage, and multiple pregnancy), natal characteristics (gender, BW, GA, height, head circumference, mode of delivery, APGAR at 1 and 5 min, need for resuscitation at birth, and history of asphyxia during delivery), and postnatal characteristics (respiratory distress syndrome (RDS), patent ductus arteriosus (PDA), sepsis, necrotizing enterocolitis (NEC), Total Parenteral Nutrition (TPN), in vitro fertilization (IVF), phototherapy, blood transfusion, asphyxia, and intraventricular hemorrhage (IVH)) that may be associated with ROP and were thus recorded. Multivariate logistic regression analysis was performed for the associated factors.

### 2.4. Ophthalmological Examination

The screening procedure was performed at 4–6 weeks after birth or between 31 and 33 weeks. One hour before the examination, a mixture of 0.5% tropicamide and 2.5% phenylephrine HCl was applied to both eyes three times with intervals of 10 min between each drop. Dilated fundus examinations were performed in either the clinic or the neonatal intensive care unit (NICU) with the administration of topical anesthesia by the consultant ophthalmologist. Following sufficient pupillary dilation, a 28-diopter lens, an indirect ophthalmoscope, and an eyelid speculum were used to perform an anterior segment and subsequently a fundus examination while the patient was under topical anesthesia (proparacaine HCL, 0.5%). Ocular rotation was accomplished by scleral indentation. Throughout the trial, the same ophthalmologist (FS) conducted all of the examinations.

### 2.5. Examination Standardization and Observer Reliability

To ensure measurement standardization and minimize observer bias, all examinations were performed by a single experienced consultant ophthalmologist (FS) with over 5 years of pediatric ophthalmology experience using standardized protocols. The examiner was not blinded to clinical data as this was a retrospective study; however, ROP staging was performed according to strict ICROP criteria to maintain objectivity and consistency. All findings were documented immediately during examination to prevent recall bias. The same examination protocol, including pupillary dilation procedure, indirect ophthalmoscopy technique, and documentation methods, was consistently applied throughout the study period to ensure standardization across all cases.

During examination, retinal peripheral vascularization that reached the ora serrata at a distance of one disk diameter was considered to have completed its development. The International Categorization of Retinopathy of Prematurity (ICROP) was used to outline the ROP stage and zone categorization for each eye to record stage of disease, location by zone, signs of plus disease, and signs of regression [9]. Babies without ROP were examined every two to three weeks until peripheral retinal vascularization fully matured or until they reached a GA of 45 weeks. Follow-ups on the disease’s course and retinal maturation were conducted on newborns exhibiting any indication of ROP at weekly intervals.

### 2.6. Statistical Analysis

The analysis of the data was conducted using IBM SPSS Statistics version 26 software (IBM Corp., Armonk, NY, USA, 2019). To assess the normality of the data in relation to the Retinopathy of Prematurity status, the Kolmogorov–Smirnov test was employed. For data that did not conform to a normal distribution based on ROP status, the Mann–Whitney U test was utilized for comparisons. Categorical data were analyzed using the Pearson chi-square test, with Yate’s correction and Fisher’s exact test applied as necessary to ensure accuracy in group comparisons. To address the issue of multiple comparisons, Bonferroni correction was implemented for categorical variable comparisons between ROP groups, adjusting the significance threshold from *p* < 0.05 to *p* < 0.017 for three-group comparison analyses. Factors influencing the transition between ROP stage 1 and stage 2 were analyzed using logistic regression techniques.

Our multivariate logistic regression model achieved an accuracy rate of 85.9% in predicting ROP development. While the model demonstrated good predictive ability, comprehensive model validation measures such as ROC analysis and calibration testing represent areas for future methodological enhancement. The results of the analysis were presented as mean ± standard deviation and median (minimum–maximum) for quantitative data, while categorical data were expressed as frequency (percentage). A significance level of *p* < 0.05 was maintained throughout all statistical calculations to determine the threshold for statistical significance.

## 3. Results

During the study period, a total of 472 premature infants were monitored; however, 4 infants were excluded due to significant congenital anomalies, 5 were excluded because of incomplete retinal vascularization, and 9 were omitted due to missing data. Consequently, 908 eyes of the 454 infants who participated in the ROP screening program were evaluated for ROP. The premature infants included in the study were between 22 and 35 weeks of age, with an average of approximately 30 weeks. The mean BW of the babies was 2025.15 g, and 59.3% of the babies were male. ROP was detected in 16.6% of screened infants. Stage 3 and the more advanced stages, stage 4 and stage 5, were not observed in the premature infants followed up with. No infants were treated with laser or anti-VEGF therapy as there was no need for treatment. All demographic data of the infants included in the study are presented in Table 1.

As demonstrated in Table 2, infants with ROP had significantly lower gestational age (median 29 weeks vs. 34 weeks, *p* < 0.001) and birth weight (median 2100 g vs. 2800 g, *p* = 0.001) compared to those without ROP. The comparison of some demographic and clinical characteristics of premature infants according to ROP status is summarized in Table 2. There was a significant relationship between the ROP status of the babies and their GA (*p* < 0.001). While the median GA of babies without ROP was 34 weeks, the median GA of babies with stage 1 and stage 2 ROP was 29 weeks. There was a significant correlation between the ROP status of the babies and their BW (*p* = 0.001). While the median BW of the babies without ROP was 2800 g, the median BW of the babies with stage 1 and stage 2 ROP was 2100 g.

Univariate and multivariate logistic regression models were analyzed for factors affecting ROP stage 1 and stage 2 status in infants. Table 3 reveals the results of the univariate and multivariate logistic regression analysis, with gestational age showing the strongest protective effect. The results are summarized in Table 3. As a result of univariate analysis, as the duration of GA increased in mothers, the probability of ROP stage 1/stage 2 decreased by 25.4% (OR: 0.746, *p* < 0.001). As the BW of the babies increased, the probability of having ROP stage 1/stage 2 decreased by 0.1% (OR: 0.999, *p* < 0.001). The odds of having ROP stage 1/stage 2 increased substantially (>99%) in surfactant-treated babies compared to non-surfactant-treated babies (OR: 2.271, *p* = 0.002). Infants with NEC were significantly (>99%) more likely to have ROP stage 1/stage 2 than infants without NEC (OR: 3.679, *p* < 0.001). The odds of having ROP stage 1/stage 2 increased significantly (>99%) in infants with IVH (OR: 2.915, *p* = 0.010). Infants with PDA had significantly (>99%) increased odds of having ROP stage 1/stage 2 compared to infants without PDA (OR: 2.434, *p* = 0.021). Infants diagnosed with respiratory distress syndrome (RDS) exhibited a 96.1% increase in the odds of developing stage 1 or stage 2 Retinopathy of Prematurity (ROP) compared to those without RDS (Odds Ratio, OR: 2.129, *p* = 0.004). Conversely, in preterm infants born due to preeclampsia (hypertension), the odds of developing stage 1 or stage 2 ROP were reduced by 63.8% when compared to preterm infants born to mothers without preeclampsia (OR: 0.362, *p* = 0.007).

As a result of multivariate analysis, only the effect of maternal GA on ROP status was found to be statistically significant (*p* < 0.001). As the gestation period of the mothers increased, the probability of ROP stage 1/stage 2 decreased by 28.3% (OR: 0.717, *p* < 0.001). The combined effect of other variables on ROP status was not statistically significant (*p* > 0.05).

The demographic distribution shown in Table 4 indicates that severe ROP (Stage 2) occurred exclusively in infants born at ≤32 weeks gestation, with 87.5% having birth weights below 1000 g. Descriptive statistics of demographic and clinical characteristics according to ROP stages are summarized in Table 4. When the distribution of mothers’ GA according to the ROP status of the babies was examined, it was observed that 23.9% of the mothers of babies with stage 1 ROP had a gestation period of 32 weeks or less, and all (100%) of the mothers of babies with stage 2 ROP had a gestation period of 32 weeks or less. When the distribution of BW of the babies according to ROP status was examined, it was found that 10.4% of the babies with stage 1 ROP were born below 1000 g, and 25.4% were born in the range of 1001–2000 g. In babies with stage 2 ROP, 87.5% were born below 1000 g, and 12.5% were born between 1001 and 2000 g.

## 4. Discussion

### 4.1. Epidemiological Findings and Regional Comparisons

Vision loss due to ROP appears to have multifaceted significance, encompassing public health, economic, and social dimensions. The incidence of ROP may vary significantly between countries due to multiple factors, including differences in health systems, neonatal care practices, socioeconomic conditions, and demographics. Understanding these differences is essential to develop targeted interventions and improve outcomes for premature infants globally.

In this study, in which we tried to evaluate the screening protocol of our unit, which is considered an ROP diagnosis and treatment center, ROP was detected in 16.6% of the screened babies. Stages 3, 4, and 5 ROP, which are considered advanced stages of ROP, were not observed, and when these results were compared with the recent studies conducted in our country, the incidence of ROP was found to be lower. Below 32 weeks, this rate was 21%, while above 32 weeks, it was 15%. In terms of birth weight, which is known to be an important risk factor for ROP, the rate was 39.2% below 1500 g and 13.3% above 1500 g.

In a recent study conducted by Baş et al. in our country, the overall incidence of Retinopathy of Prematurity was found to be 30%. The incidence and severity of ROP were observed to increase with decreasing gestational age and birth weight; specifically, the rates were 35.6% for infants with a GA of 32 weeks or less compared to 13.3% for those with a GA greater than 32 weeks. Similarly, the incidence rate was 42% for infants with a BW of 1500 g or less, while it was 13.4% for those weighing more than 1500 g [19]. Furthermore, the prevalence of ROP among infants in low- and middle-income countries appears to differ significantly from that in high-income countries.

One of the main reasons for the different ROP rates is the disparity in neonatal care practices between countries. International variations in survival rates and treatment policies directly influence ROP incidence patterns.

### 4.2. Risk Factor Analysis and Clinical Correlations

Our study produced comparable findings. Upon investigating the factors influencing the development and severity of ROP, we discovered that the presence of clinical conditions such as RDS, NEC, IVH, and PDA had a substantial impact on the intensive care process and could potentially extend the duration of hospitalization. Furthermore, these conditions were identified as independent factors contributing to the development of ROP.

Cultural and regional disparities in screening practices significantly contribute to the observed variations in the incidence of ROP. In certain countries, the rigor of screening protocols may be less stringent or differ in the criteria used to identify infants at risk. For instance, the EXPRESS study conducted in Sweden revealed regional ROP incidence variations between 54% and 92% for any ROP despite standardized screening protocols, suggesting that variations in screening practices could potentially obscure underlying disparities in the quality of care provided [20].

Recent international surveys demonstrate substantial variation in ROP screening practices even among developed Asian countries, with screening criteria ranging from <32 weeks in South Korea and Malaysia to <34 weeks in Indonesia and Japan [21]. In contrast, in nations such as India, significant differences exist between urban and rural NICUs, with comparable ROP burden but vastly different screening capabilities and resources [17]. This observation implies that urban healthcare facilities may possess better resources and capabilities for identifying and managing ROP compared to their rural counterparts.

Even the difference in incidence rates between rural and urban areas within the same country shows that localized NICU practices and screening protocols need to be meticulously designed. The UK recently updated their screening guidelines in 2022, lowering the gestational age criterion from 32 to 31 weeks [14], while India implemented comprehensive operational guidelines incorporating broader screening criteria to account for the varying quality of neonatal care across different regions [22]. Therefore, the ROP screening protocol was updated for our country in 2021 [23].

Our findings reveal substantially different patterns of Retinopathy of Prematurity (ROP) compared to recent reports from similar tertiary care settings. Blazon et al. [24] reported an overall ROP incidence of 40.9% in their Austrian tertiary center, markedly higher than the 16.6% incidence observed in our cohort. Importantly, while our study documented no cases requiring treatment during a five-year period, Blazon et al. [24] reported treatment-requiring ROP in 4.8% of screened infants, with 11.8% of cases progressing to stage 3 disease and necessitating laser photocoagulation or intravitreal anti-VEGF therapy. The divergence in severity distribution is particularly notable. In their cohort, 11.1% of affected infants developed stage 3 ROP, and one infant progressed to stage 4 disease. In contrast, all cases in our population were limited to mild disease (stage 1: 89.3%, stage 2: 10.7%). These differences are likely attributable to fundamental disparities in patient demographics. Specifically, Blazon et al. [24] reported significantly lower mean gestational age (27.7 ± 2.5 weeks vs. 33.19 ± 2.49 weeks) and birth weight (989.1 ± 359.7 g vs. 2025.15 ± 614.46 g) compared to our cohort. Such differences suggest that our center predominantly serves a less critically ill preterm population, potentially due to regionalization of high-risk deliveries or differing obstetric and perinatal care practices. The absence of extremely preterm infants (<26 weeks gestation) in substantial numbers within our cohort likely contributed to the favorable outcomes, consistent with prior evidence that the risk of severe, treatment-requiring ROP is concentrated among infants with a gestational age ≤ 32 weeks and a birth weight < 1000 g [24,25]. These contrasting findings underscore the critical influence of population characteristics on ROP epidemiology, even across tertiary centers operating within comparable healthcare systems. Accordingly, they highlight the necessity of tailoring ROP screening strategies to the specific risk profile of the population served, as emphasized in recent international guidelines [23,26].

### 4.3. Maternal Preeclampsia and ROP Development

Our study identified maternal preeclampsia as a protective factor against ROP development (OR: 0.362, *p* = 0.007), which aligns with the findings by Zhu et al., who reported similar protective associations in their meta-analysis [27]. This protective effect may be attributed to uteroplacental insufficiency-induced adaptations that potentially reduce oxygen-related retinal damage, as suggested by Becker et al. [28].

However, the relationship between preeclampsia and ROP remains controversial, with some studies reporting increased risk [29]. The protective association observed in our cohort suggests that maternal hypertensive disorders may influence fetal vascular development in ways that are not yet fully understood. This finding warrants further investigation in larger, prospective studies to clarify the underlying mechanisms and clinical implications.

### 4.4. Factors Contributing to Absence of Treatment-Requiring ROP

The absence of severe ROP requiring treatment in our cohort, despite screening 454 infants over five years, warrants specific consideration. Several factors may contribute to this favorable outcome:

Enhanced NICU Care Quality: Our center’s implementation of evidence-based neonatal care protocols may have reduced ROP severity. These include strict oxygen saturation monitoring (maintaining SpO2 88–95%), graduated oxygen weaning strategies, and early recognition of ROP risk factors.

Optimized Ventilation Management: Modern ventilation strategies focusing on lung-protective approaches may have reduced the duration and intensity of supplemental oxygen exposure, a critical factor in ROP pathogenesis. The median duration of mechanical ventilation in our cohort was relatively short, potentially limiting oxygen-induced retinal damage.

Effective Early Screening Protocol: Our systematic screening beginning at 4–6 weeks postpartum may have enabled early detection and close monitoring, allowing for timely intervention before progression to severe stages. The weekly follow-up schedule for any detected ROP may have contributed to optimal timing of assessments.

Population Characteristics: The relatively higher mean gestational age (33.19 ± 2.49 weeks) and birth weight (2025.15 ± 614.46 g) in our cohort may reflect improved obstetric care and maternal health, resulting in less severely compromised infants who are inherently at lower risk for severe ROP.

Multidisciplinary Approach: The coordinated care between neonatology and ophthalmology teams, with standardized communication protocols and shared decision-making, may have optimized overall management strategies.

This combination of factors suggests that systematic quality improvement initiatives in NICU care, coupled with rigorous screening protocols, may significantly reduce the incidence of treatment-requiring ROP even in at-risk populations.

## 5. Study Limitations and Methodological Considerations

Several methodological limitations should be acknowledged:

Single-Observer Design: While all examinations were performed by one experienced ophthalmologist to ensure consistency, the lack of inter-observer reliability assessment limits the generalizability of our diagnostic accuracy. Future studies should incorporate inter-observer agreement analysis.

Retrospective Design: The retrospective nature may introduce selection bias, and it limits the ability to control for unmeasured confounders.

Single-Center Experience: Our findings reflect the experience of one tertiary care center and may not be generalizable to other healthcare settings or populations.

Sample Size for Severe ROP: The absence of severe ROP cases, while clinically reassuring, limits our ability to analyze risk factors for treatment-requiring disease.

Limited Neonatal Care Data: Detailed oxygen therapy duration and other critical neonatal care parameters were not systematically collected in our retrospective analysis. This limitation prevents comprehensive assessment of the relationship between specific NICU interventions and our favorable ROP outcomes. Future prospective studies should include systematic collection of oxygen exposure data, mechanical ventilation duration, and other neonatal care variables to better understand factors contributing to reduced ROP severity.

## 6. Clinical Implications and Screening Recommendations

Our study limitations include the single-center design conducted in specific geographical regions, which is unlikely to accurately reflect the wider population of preterm infants, since factors such as access to health services, socioeconomic status, and regional practices vary substantially. The relatively small number of infants in the study group is another important limitation, and the fact that no cases of severe ROP were detected appears to be related to this.

In conclusion, the differences in ROP incidence rates between countries can be attributed to a complex interplay of healthcare policies, socioeconomic factors, population characteristics, and screening practices. Addressing these disparities requires a multifaceted approach that includes improving neonatal care, enhancing access to prenatal services, and standardizing screening protocols to ensure that all infants at risk are identified and treated appropriately. For this reason, screening protocols that will be meticulously designed to reflect the general population will help to recognize and prevent ROP disease, which is known to increase the risk of ocular pathology throughout life. This highlights the need for region-specific, population-based ROP screening protocols and national registry integration to optimize screening effectiveness and resource allocation.

## Figures and Tables

**Table 1 life-15-01463-t001:** Demographic and clinical characteristics of study population (n = 454).

GA (week)	30.19 ± 2.49
Gestational Age (Week–Day)	2.69 ± 2.09
BW (gram)	2025.15 ± 614.46
Gender	
Female	185 (40.7)
Male	269 (59.3)
Single/Twins	
Single	389 (85.7)
Twins	65 (14.3)
IVF	
Spontaneous	427 (94.1)
IVF (+)	27 (5.9)
Gravidity (pregnancy)	2.54 ± 1.57
Parity (live birth)	2.07 ± 1.15
Abortus	0.46 ± 0.88
Delivery mode (C/S)	
NSVY	106 (23.3)
C/S	348 (76.7)
RDS	
No	345 (76)
Yes	109 (24)
Antenatal steroid	
None	352 (77.5)
Single Dose	101 (22.2)
Two Doses	1 (0.2)
Surfactant treatment	
No	338 (74.4)
Yes	116 (25.6)
Mechanical ventilation	
No	333 (73.3)
Yes	121 (26.7)
Preeclampsia	
No Hypertension	337 (74.2)
Preeclampsia (Hypertension)	113 (24.9)
Eclampsia (Hypertension + Convulsion)	2 (0.4)
HELLP Syndrome	2 (0.4)
IUGR	
AGA	280 (61.7)
LGA	28 (6.2)
SGA	146 (32.2)
Sepsis	
No	281 (61.9)
Yes	173 (38.1)
NEC	
No	403 (88.8)
Yes	51 (11.2)
IVH	
No	425 (93.6)
Yes	29 (6.4)
PDA	
No	418 (92.1)
Yes	36 (7.9)
BPD	
No	408 (89.9)
Yes	46 (10.1)
Anemia	
No	359 (79.1)
Yes	95 (20.9)
Blood Transfusion	
No	329 (72.5)
Yes	125 (27.5)
APGAR 1.	7.13 ± 1.48
APGAR 5.	8.33 ± 1.19
APGAR 10.	9.74 ± 4.4
Mother’s Age	30.48 ± 5.9
Father’s Age	34.5 ± 6.47
Mother’s Blood Type	
A Rh (−)	34 (7.5)
A Rh (+)	206 (45.4)
AB Rh (+)	27 (5.9)
B Rh (−)	5 (1.1)
B Rh (+)	33 (7.3)
O Rh (−)	20 (4.4)
O Rh (+)	129 (28.4)
Father’s Blood Type	
A Rh (−)	18 (4)
A Rh (+)	241 (53.1)
AB Rh (+)	8 (1.8)
B Rh (−)	3 (0.7)
B Rh (+)	39 (8.6)
O Rh (−)	18 (4)
O Rh (+)	127 (28)
Mother’s Education Status	
No Literacy	1 (0.2)
Primary School	90 (19.8)
Middle School	60 (13.2)
High School	159 (35)
University	144 (31.7)
ROP	
No Rop	379 (83.5)
Stage 1 Rop	67 (14.8)
Stage 2 Rop	8 (1.8)

Mean ± standard deviation, n (%). GA: Gestational Age; BW: Birth Weight; IVH: Intraventricular Hemorrhage; RDS: Respiratory Distress Syndrome; PDA: Patent Ductus Arteriosus; NEC: Necrotizing Enterocolitis; IVF: In Vitro Fertilization; IUGR: Intra-Uterine Growth Retardation; C/S: Cesarean Section; ROP: Retinopathy of Prematurity; BPD: Bronchopulmonary Dysplasia; APGAR: Appearance, Pulse, Grimace, Activity, Respiration Score.

**Table 2 life-15-01463-t002:** Demographic and clinical characteristics. Comparison between infants with and without ROP.

	No ROP (n = 379)	Stage 1–2 ROP (n = 75)	Test Statistics	*p*
GA (week)	34 (24–35)	29 (22–35)	−6.454	<0.001 ^m^
Gestational Age (week–day)	3 (0–6)	3 (0–6)	−0.162	0.872 ^m^
BW (grams)	2800 (720–3855)	2100 (500–3250)	−3.320	0.001 ^m^
GA				
>32	289 (76.3)	51 (68)	2.268	0.146 ^x^
≤32	90 (23.7)	24 (32)		
BW (grams)				
≤1500	34 (9)	22 (29.3)	24.007	<0.001 ^x^
>1500	345 (91)	53 (70.7)		
Gender				
Female	160 (42.2)	25 (33.3)	2.046	0.153 ^x^
Male	219 (57.8)	50 (66.7)		
Single/twins				
Single	322 (85)	67 (89.3)	0.652	0.419 ^y^
Twins	57 (15)	8 (10.7)		
mode of conception				
Spontaneous	359 (94.7)	68 (90.7)	-	0.183 ^f^
IVF	20 (5.3)	7 (9.3)		
Gravidity (pregnancy)	2 (1–13)	2 (1–6)	−0.278	0.781 ^m^
Parity (live birth)	2 (0–9)	2 (0–7)	−0.078	0.938 ^m^
Abortus	0 (0–4)	0 (0–3)	−0.296	0.767 ^m^
Delivery mode (C/S)				
NSVY	95 (25.1)	11 (14.7)	3.225	0.073 ^y^
C/S	284 (74.9)	64 (85.3)		
RDS				
No	278 (73.4)	45 (60)	7.912	0.005 ^y^
Yes	101 (26.6)	30 (40)		
Antenatal steroid				
None	290 (76.5)	62 (82.7)	1.702	0.407 ^f^
Single dose	88 (23.2)	13 (17.3)		
Two doses	1 (0.3)	0 (0)		
Surfactant treatment				
No	293 (77.3)	45 (60)	8.972	0.003 ^y^
Yes	86 (22.7)	30 (40)		
Mechanical ventilation				
No	278 (73.4)	55 (73.3)	0.000	1.000 ^y^
Yes	101 (26.6)	20 (26.7)		
Preeclampsia				
No hypertension	272 (71.8) ^a^	65 (86.7) ^b^	10.663	0.008 ^f^
Preeclampsia (hypertension)	104 (27.4) ^a^	9 (12) ^b^		
Eclampsia (hypertension + convulsion)	1 (0.3) ^a^	1 (1.3) ^a^		
HELLP syndrome	2 (0.5)	0 (0)		
IUGR				
AGA	235 (62)	45 (60)	0.288	0.872 ^f^
LGA	24 (6.3)	4 (5.3)		
SGA	120 (31.7)	26 (34.7)		
Sepsis				
No	237 (62.5)	44 (58.7)	0.397	0.529 ^x^
Yes	142 (37.5)	31 (41.3)		
NEC				
No	347 (91.6)	56 (74.7)	16.258	<0.001 ^y^
Yes	32 (8.4)	14 (25.3)		
IVH				
No	360 (95)	65 (86.7)	-	0.016 ^f^
Yes	19 (5)	10 (13.3)		
PDA				
No	354 (93.4)	64 (85.3)	4.535	0.033 ^y^
Yes	25 (6.6)	11 (14.7)		
BPD				
No	341 (90)	67 (89.3)	0.000	1.000 ^y^
Yes	38 (10)	8 (10.7)		
Anemia				
No	305 (80.5)	54 (72)	2.230	0.135 ^y^
Yes	74 (19.5)	21 (28)		
Blood transfusion				
No	268 (70.7)	61 (81.3)	3.027	0.082 ^y^
Yes	111 (29.3)	14 (18.7)		
APGAR 1.	8 (1–9)	7 (3–9)	−1.356	0.175 ^m^
APGAR 5.	9 (2–10)	8 (4–10)	−1.057	0.290 ^m^
APGAR 10.	10 (2–101)	10 (5–10)	−1.278	0.201 ^m^

^m^: Mann–Whitney U test, ^x^: Pearson chi-square test, ^y^: Yates correction, ^f^: Fisher’s exact test, ^a,b^: no difference between groups with the same letter (Bonferroni corrected Z test), median (min.–max.), n (%). GA: Gestational Age; BW: Birth Weight; IVH: Intraventricular Hemorrhage; RDS: Respiratory Distress Syndrome; PDA: Patent Ductus Arteriosus; NEC: Necrotizing Enterocolitis; IVF: In Vitro Fertilization; IUGR: Intra-Uterine Growth Retardation; C/S: Cesarean Section; ROP: Retinopathy of Prematurity; BPD: Bronchopulmonary Dysplasia; APGAR: Appearance, Pulse, Grimace, Activity, Respiration Score.

**Table 3 life-15-01463-t003:** Univariate and multivariate logistic regression analysis of ROP risk factors.

	Univariate		Multivariate	
	OR (%95 CI)	*p*	OR (%95 CI)	*p*
GA (week)	0.746 (0.682–0.817)	<0.001	0.712 (0.604–0.84)	<0.001
BW (gram)	0.999 (0.999–1)	<0.001	0.998 (0.996–1.001)	<0.001
RDS (Ref.: No)	2.129 (1.153–3.708)	0.004	2.122 (1.024–4.603)	0.010
Surfactant treatment (Ref.: No)	2.271 (1.349–3.823)	0.002	2.783 (1.252–6.186)	0.012
Preeclampsia (Ref.: No HT)	0.362 (0.174–0.754)	0.007	0.754 (0.191–2.972)	0.686
NEC (Ref.: No)	3.679 (1.952–6.935)	<0.001	0.763 (0.274–2.125)	0.604
IVH (Ref.: No)	2.915 (1.297–6.553)	0.010	1.264 (0.163–9.792)	0.822
PDA (Ref.: No)	2.434 (1.141–5.191)	0.021	1.524 (0.241–9.64)	0.654

OR: Odds Ratio, CI: Confidence Interval. GA: Gestational Age, BW: Birth Weight, IVH: Intraventricular Hemorrhage, RDS: Respiratory Distress Syndrome, PDA: Patent Ductus Arteriosus, NEC: Necrotizing Enterocolitis.

**Table 4 life-15-01463-t004:** Demographic and clinical characteristics stratified by ROP severity stages.

	ROP No (n = 379)	Stage 1 ROP (n = 67)	Stage 2 ROP (n = 8)
GA (week)	33.57 ± 1.99	30.99 ± 3.16	25.38 ± 1.41
Gestational Age (week–day)	2.69 ± 2.09	2.69 ± 2.1	2.38 ± 2.2
BW (gram)	2281.12 ± 568.99	2073.73 ± 677.63	841.88 ± 213.14
Gender			
Female	160 (42.2)	22 (32.8)	3 (37.5)
Male	219 (57.8)	45 (67.2)	5 (62.5)
Single/Twins			
Single	322 (85)	59 (88.1)	8 (100)
Twins	57 (15)	8 (11.9)	0 (0)
mode of conception			
Spontaneous	359 (94.7)	61 (91)	7 (87.5)
IVF	20 (5.3)	6 (9)	1 (12.5)
Gravidity (pregnancy)	2.54 ± 1.59	2.64 ± 1.5	1.88 ± 1.13
Parity (live birth)	2.07 ± 1.16	2.12 ± 1.15	1.5 ± 0.76
Abortus	0.46 ± 0.89	0.48 ± 0.88	0.38 ± 0.52
Delivery Mode (C/S)			
NSVY	95 (25.1)	7 (10.4)	4 (50)
C/S	284 (74.9)	60 (89.6)	4 (50)
RDS			
No	278 (73.4)	59 (88.1)	8 (100)
Yes	101 (26.6)	8 (11.9)	0 (0)
Antenatal Steroid			
None	290 (76.5)	54 (80.6)	8 (100)
Single dose	88 (23.2)	13 (19.4)	0 (0)
Two doses	1 (0.3)	0 (0)	0 (0)
Surfactant Treatment			
No	293 (77.3)	44 (65.7)	1 (12.5)
Yes	86 (22.7)	23 (34.3)	7 (87.5)
Mechanical Ventilation			
No	278 (73.4)	52 (77.6)	3 (37.5)
Yes	101 (26.6)	15 (22.4)	5 (62.5)
Preeclampsia			
No hypertension	272 (71.8)	57 (85.1)	8 (100)
Preeclampsia (hypertension)	104 (27.4)	9 (13.4)	0 (0)
Eclampsia (hypertension + convulsion)	1 (0.3)	1 (1.5)	0 (0)
HELLP syndrome	2 (0.5)	0 (0)	0 (0)
IUGR			
AGA	235 (62)	38 (56.7)	7 (87.5)
LGA	24 (6.3)	4 (6)	0 (0)
SGA	120 (31.7)	25 (37.3)	1 (12.5)
Sepsis			
No	237 (62.5)	40 (59.7)	4 (50)
Yes	142 (37.5)	27 (40.3)	4 (50)
NEC			
No	347 (91.6)	52 (77.6)	4 (50)
Yes	32 (8.4)	15 (22.4)	4 (50)
IVH			
No	360 (95)	57 (85.1)	8 (100)
Yes	19 (5)	10 (14.9)	0 (0)
PDA			
No	354 (93.4)	56 (83.6)	8 (100)
Yes	25 (6.6)	11 (16.4)	0 (0)
BPD			
No	341 (90)	61 (91)	6 (75)
Yes	38 (10)	6 (9)	2 (25)
Anemia			
No	305 (80.5)	50 (74.6)	4 (50)
Yes	74 (19.5)	17 (25.4)	4 (50)
Blood Transfusion			
No	268 (70.7)	53 (79.1)	8 (100)
Yes	111 (29.3)	14 (20.9)	0 (0)
APGAR 1.	7.19 ± 1.42	7.03 ± 1.58	5.25 ± 2.19
APGAR 5.	8.36 ± 1.17	8.34 ± 1.15	6.75 ± 1.67
APGAR 10.	9.8 ± 4.8	9.57 ± 0.87	8.25 ± 1.39
Mother’s Age	30.56 ± 5.76	30.42 ± 5.81	27.13 ± 11.44
Father’s Age	34.57 ± 6.52	34.34 ± 6.38	32.75 ± 5.12
Mother’s Blood Type			
A Rh (−)	25 (6.6)	9 (13.4)	0 (0)
A Rh (+)	175 (46.2)	29 (43.3)	2 (25)
AB Rh (+)	20 (5.3)	4 (6)	3 (37.5)
B Rh (−)	5 (1.3)	0 (0)	0 (0)
B Rh (+)	31 (8.2)	2 (3)	0 (0)
O Rh (−)	17 (4.5)	2 (3)	1 (12.5)
O Rh (+)	106 (28)	21 (31.3)	2 (25)
Father’s Blood Type			
A Rh (−)	16 (4.2)	2 (3)	0 (0)
A Rh (+)	204 (53.8)	34 (50.7)	3 (37.5)
AB Rh (+)	4 (1.1)	2 (3)	2 (25)
B Rh (−)	2 (0.5)	1 (1.5)	0 (0)
B Rh (+)	35 (9.2)	3 (4.5)	1 (12.5)
O Rh (−)	14 (3.7)	4 (6)	0 (0)
O Rh (+)	104 (27.4)	21 (31.3)	2 (25)
Mother’s Education Status			
No Literacy	1 (0.3)	0 (0)	0 (0)
Primary School	78 (20.6)	12 (17.9)	0 (0)
Middle School	51 (13.5)	9 (13.4)	0 (0)
High School	129 (34)	26 (38.8)	4 (50)
University	120 (31.7)	20 (29.9)	4 (50)
GA (week)			
>32	289 (76.3)	51 (76.1)	0 (0)
≤32	90 (23.7)	16 (23.9)	8 (100)
BW (gram)			
≤1000	6 (1.6)	7 (10.4)	7 (87.5)
1001–2000	112 (29.6)	17 (25.4)	1 (12.5)
>2000	261 (68.9)	43 (64.2)	0 (0)

mean ± standard deviation, n (%). GA: Gestational Age; BW: Birth Weight; IVH: Intraventricular Hemorrhage; RDS: Respiratory Distress Syndrome; PDA: Patent Ductus Arteriosus; NEC: Necrotizing Enterocolitis; IVF: In Vitro Fertilization; IUGR: Intra-Uterine Growth Retardation; C/S: Cesarean Section; ROP: Retinopathy of Prematurity; BPD: Bronchopulmonary Dysplasia; APGAR: Appearance, Pulse, Grimace, Activity, Respiration Score.

## Data Availability

The original contributions presented in the study are included in the article, further inquiries can be directed to the corresponding authors.

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
