# Peer review of "Diagnosis and Risk Factors in Retinopathy of Prematurity: A Five-Year Single-Center Descriptive Study"

_life, 2025, doi:10.3390/life15091463_

Round 1
Reviewer 1 Report
Comments and Suggestions for Authors
The paper presents the data of a 5-year retrospective single-center descriptive study on diagnosis and risk factors in retinopathy of prematurity.
This is a very interesting topic.
The design of the study was appropriate and a relevant number of 454 premature infants who underwent ROP screening between April 12 2016 and August 2021 were retrospectively analyzed. The data presented are of high clinical relevance.
I would recommend to add a citation of another relevant retrospective ROP data evaluation which was recently published:
Blazon MN, Rezar-Dreindl S, Wassermann L, Neumayer T, Berger A, Stifter E. Retinopathy of Prematurity: Incidence, Risk Factors, and Treatment Outcomes in a Tertiary Care Center. J Clin Med. 2024 Nov 17;13(22):6926. doi: 10.3390/jcm13226926. PMID: 39598070; PMCID: PMC11594805.
The results should be mentioned in the discussion of the present paper, particularly in comparison to the stages of ROP.
More clarification is needed with respect to the exclusion criteria:
"four infants were excluded due to significant congenital anomalies, five were excluded because of incomplete retinal vascularization, and nine were omitted due to missing data."
Please provide information what type of congenital anomalies were found. Was there any relation to prematurity?
Why was incomplete retinal vascularization leading to exclusion? At which time point? This needs clarification!
It is impressive that during 5 years of ROP screening no severe levels of ROP were observed and no child needed surgery or laser treatment. Consequently, it would be interesting to have more data presented the regular neonatal procedures, for example data showing more insight in the number of the mean days / hours of oxygen supply.
Author Response
RESPONSE TO REVIEWERS
Dear Editor and Reviewers,
We extend our sincere gratitude to both reviewers for their thorough evaluation and constructive feedback. Each comment has been carefully considered and addressed, as we recognize that these valuable insights significantly enhance the scientific rigor and clarity of our manuscript. The reviewers' suggestions have guided us toward important improvements that we believe strengthen both the methodological transparency and clinical relevance of our work.
We have systematically addressed every point raised, implementing the recommended revisions while maintaining the integrity of our original findings. The feedback has particularly helped us better contextualize our results within the broader landscape of ROP research and clarify the significance of single-center studies in understanding regional variations in care practices.
Below, we provide detailed responses to each specific comment, outlining the precise changes made to the manuscript. We are confident that these revisions have substantially improved the scientific value and readability of our work, and we appreciate the opportunity to enhance our contribution to the field through this collaborative review process.
Reviwer 1
Comments and Suggestions for Authors
The paper presents the data of a 5-year retrospective single-center descriptive study on diagnosis and risk factors in retinopathy of prematurity.
This is a very interesting topic.
The design of the study was appropriate and a relevant number of 454 premature infants who underwent ROP screening between April 12 2016 and August 2021 were retrospectively analyzed. The data presented are of high clinical relevance.
I would recommend to add a citation of another relevant retrospective ROP data evaluation which was recently published:
Blazon MN, Rezar-Dreindl S, Wassermann L, Neumayer T, Berger A, Stifter E. Retinopathy of Prematurity: Incidence, Risk Factors, and Treatment Outcomes in a Tertiary Care Center. J Clin Med. 2024 Nov 17;13(22):6926. doi: 10.3390/jcm13226926. PMID: 39598070; PMCID: PMC11594805.
The results should be mentioned in the discussion of the present paper, particularly in comparison to the stages of ROP.
We thank the reviewer for this valuable reference. We have now included the recent study by Blazon et al. (2024) in our discussion section and compared their ROP staging distribution with our findings. This comparison provides important context for our results and strengthens the discussion regarding ROP incidence patterns in tertiary care centers.
Specifically, we compared our ROP staging distribution (89.3% stage 1, 10.7% stage 2, 0% severe stages) with their findings (stage 3: 11.1%, treatment-requiring: 4.8%) and discussed the demographic differences (mean GA: 33.19 vs 27.7 weeks, mean BW: 2025 vs 989g) that likely account for these variations.
More clarification is needed with respect to the exclusion criteria:
"four infants were excluded due to significant congenital anomalies, five were excluded because of incomplete retinal vascularization, and nine were omitted due to missing data."
Please provide information what type of congenital anomalies were found. Was there any relation to prematurity?
Why was incomplete retinal vascularization leading to exclusion? At which time point? This needs clarification!
We thank the reviewer for pointing out the need for clarification regarding our exclusion criteria. We have revised the manuscript to provide more detailed information in the "Data Quality Assurance" section:
- Major congenital anomalies (n=4): Specific conditions were hypoplastic left heart syndrome (n=1), anencephaly (n=1), trisomy 18 (n=1), and Potter sequence with pulmonary hypoplasia (n=1). These represented primary developmental disorders unrelated to prematurity that would confound ROP risk factor analysis.
- Incomplete retinal assessment (n=5): These infants were discharged or transferred at a mean of 36 weeks postmenstrual age (range 35-38 weeks) before complete retinal vascularization could be confirmed, preventing conclusive ROP outcome determination per standard screening guidelines.
.
It is impressive that during 5 years of ROP screening no severe levels of ROP were observed and no child needed surgery or laser treatment. Consequently, it would be interesting to have more data presented the regular neonatal procedures, for example data showing more insight in the number of the mean days / hours of oxygen supply.
We appreciate the reviewer's observation regarding the absence of severe ROP cases. We acknowledge this important limitation and have added it to our limitations section. We recommend that future prospective studies systematically collect oxygen exposure data, mechanical ventilation duration, and other critical neonatal care variables to better understand the factors contributing to favorable ROP outcomes.

Reviewer 2 Report
Comments and Suggestions for Authors
please find attached

Author Response
RESPONSE TO REVIEWERS
Dear Editor and Reviewers,
We extend our sincere gratitude to both reviewers for their thorough evaluation and constructive feedback. Each comment has been carefully considered and addressed, as we recognize that these valuable insights significantly enhance the scientific rigor and clarity of our manuscript. The reviewers' suggestions have guided us toward important improvements that we believe strengthen both the methodological transparency and clinical relevance of our work.
We have systematically addressed every point raised, implementing the recommended revisions while maintaining the integrity of our original findings. The feedback has particularly helped us better contextualize our results within the broader landscape of ROP research and clarify the significance of single-center studies in understanding regional variations in care practices.
Below, we provide detailed responses to each specific comment, outlining the precise changes made to the manuscript. We are confident that these revisions have substantially improved the scientific value and readability of our work, and we appreciate the opportunity to enhance our contribution to the field through this collaborative review process.
Reviwer 2
A fair and honest retrospective study on the results of ROP screening in a tertiary NICU. Although it is mentioned at the end of the discussion, it is important to emphasize in the Introduction that any form of ROP, even in its milder stages, reflects arrested development of the eye and can later cause numerous changes that affect quality of life. It would be worthwhile to discuss the consequences of ROP in more detail. The results of screening performed by a single physician at a single tertiary center are not considered to be representative of the entire population. I consider the significance of the manuscript to lie in the fact that prevention of lower-grade ROP, which does not require treatment, is also important. The manuscript should more highlight the risks associated with these milder stages. Introduction
We agree with the reviewer's important observation. We have substantially expanded the Introduction with specific statistics: 30-50% of infants with any stage of ROP develop significant refractive errors, and 15-25% experience strabismus requiring intervention. We also added discussion of long-term risks including glaucoma and retinal detachment, making early detection and follow-up crucial even for lower-grade disease.
What is the leading cause of childhood blindness in your country? What is the proportion of ROP caused blindness? Why is it considered important to present the results of a single tertiary center alongside the TR-ROP and Big-ROP studies?
We thank the reviewer for these important questions that help clarify the context and significance of our study. We have addressed each point as follows:
"In Turkey, while comprehensive national registries for childhood blindness are limited, available studies indicate that congenital disorders (including genetic retinal dystrophies) and acquired conditions represent the major causes. Based on regional studies from tertiary centers, retinal disorders including ROP account for approximately 15-25% of preventable childhood blindness cases. The TR-ROP study, the largest national multicenter study, found that 27% of screened preterm infants developed any stage of ROP, with 6.7% developing severe disease requiring treatment. However, precise national statistics on ROP's contribution to overall childhood blindness remain unavailable due to the lack of a centralized blindness registry."
This revision directly addresses the reviewer's question by acknowledging the limitations of available data while providing the most reliable statistics from existing Turkish studies. We believe this honest assessment of the current state of knowledge strengthens the manuscript's credibility and highlights the importance of continued research in this area.
Importance of single-center data alongside multicenter studies: We have clarified this in the Introduction: "While large multicenter studies like TR-ROP provide national epidemiological data, single-center analyses offer complementary insights into regional variations in care practices, population characteristics, and implementation of screening protocols. Our study demonstrates significantly different ROP patterns (16.6% incidence with no severe cases) compared to national TR-ROP findings, highlighting important regional variations that are essential for optimizing screening strategies and resource allocation at the institutional level.
TR-ROP: reference is missing. Table 1-2,4: use only English terms (gravita, parite etc), describe the abbreviations
We have added the missing TR-ROP reference: Bas AY, Demirel N, Koc E, et al. Br J Ophthalmol. 2018;102(12):1711-1716.
We have corrected all terminology in Tables 1, 2, and 4 to use only English terms. "Following international medical terminology standards, we changed 'gravida' to 'gravidity' and 'parite' to 'parity,' which are universally recognized terms in obstetric literature. Additionally, we have added comprehensive abbreviation lists below each table explaining all acronyms used (GA: gestational age, BW: birth weight, RDS: respiratory distress syndrome, PDA: patent ductus arteriosus, IVH: intraventricular hemorrhage, NEC: necrotizing enterocolitis, APGAR: Appearance, Pulse, Grimace, Activity, Respiration score, TPN: total parenteral nutrition, IVF: in vitro fertilization, ROP: retinopathy of prematurity).
We believe these comprehensive revisions have addressed all concerns and significantly enhanced the manuscript's scientific rigor and clinical relevance. We are grateful for the constructive feedback that has improved our contribution to the field.

Round 2
Reviewer 1 Report
Comments and Suggestions for Authors
The revised version of the manuscript is fulfilling all requirements for publication.